# Leader extraversion and team performance: A moderated mediation model

**Jun Zhang**[1]*, **Kui Yin**[2], **SiQi Li**[1]

**1** Faculty of Humanities and Social Sciences, Beijing Institute of Petrochemical Technology, Beijing, People's Republic of China, **2** Donlinks School of Economics and Management, University of Science and Technology Beijing, Beijing, People's Republic of China

* zhangj@outlook.com

**Data Availability Statement:** All relevant data are within the paper and its Supporting information files.

**Funding:** This study was supported by R&D Program of Beijing Municipal Education Commission(SM202110017003) awarded to the first author (Jun Zhang). URL of the funder

## Abstract

Extraversion is the best and most consistent predictor of important leadership outcomes. However, there has been little exploration and examination of the mechanisms underlying the effects of extraverted leadership on performance. Drawing on distal-proximal motivational theory and situational strength theory, the present study proposes and examines a moderated mediation model that explains how leader extraversion affects team performance and how situational characteristics strengthen or constrain this relationship. Respondents were recruited through management team training courses run by the eight Chinese companies. We conducted two rounds of electronic questionnaire collection. The first round of data was collected during the training session. Four weeks later, we collected the data through the training courses' WeChat groups. Data collected from 226 Chinese team leaders was analyzed using SPSS 26 and Mplus 7. We find that leader extraversion predicts team performance through a motivational mechanism operationalized as leader work engagement. We further find that goal clarity and process clarity play an important role in strengthening the positive effect of leader extraversion on leader work engagement as well as the motivational mechanism, providing an empirical explanation of how leader extraversion affects team performance through a motivational mechanism operationalized as leader work engagement. We also explore how two potential situational characteristics, operationalized as goal clarity and process clarity of leaders, affect the relationship between leader extraversion and leader work engagement as well as the motivational mechanism. Addionally, the findings suggest important practical implications for the organizations seeking to identify effective team leaders.

## Introduction

Extraverted personality has long been acclaimed as the most effective personality variable for predicting leadership [1]. Within organizations, leaders influence others through interactions to motivate them to reach team and organizational goals. This requires leaders to possess characteristics consistent with extraversion: sociablility, contagiours energy, assertiveness, and gregariousness [2–5]. Two seminal meta-analytic research papers [1, 6] have established that of

website: http://jw.beijing.gov.cn/kyc/ The funders had no role in study design, data collection and analysis, decision to publish, or preparation of the manuscript.

**Competing interests:** The authors have declared that no competing interests exist.

the five-factor model (FFM) of personality traits, extraversion is the best and most consistent predictor of important leadership outcomes (i.e., leader emergence, leadership effectiveness, and transformational leadership). Furthermore, one most recent meta-analysis [7] focusing solely on extraversion provides more detailed and in-depth findings showing that extraversion and its lower-order traits are consistent predictors of leader emergence, leader behaviors (for both transformational and transactional leadership), and leadership effectiveness. Compared to other traits, thus, extraverted personality has an advantage in predicting leadership and performance.

Despite its obvious advantages for working environments, there has been limited research to date on the mechanisms underlying the effects of extraverted leadership on performance [8]. Research on how extraverted leadership affects team performance and the circumstances in which the effects of extraverted leadership are inhibited or enhanced remains limited. Judge et al. [1] called for further research to focus on processes of influence of leadership traits and contexts in which leadership traits have an effect. In particular, the mechanisms of influence of the Big Five personality traits are directly related to the entire field of leadership traits research [1, 4]. This article, therefore, explores the process of influencing team performance and its contextual variables in terms of extraverted personality.

In this article, we draw on earlier work on the associations between leader extraversion and team performance and proposed our hypotheses from a theoretical perspective that was different from those prior works. Drawing on the distal-proximal motivation theory [9] and the situational strength theory [10], we develop a moderated mediation model (Fig 1) that helps explain both how leader extraversion affects team performance and when such effects are suppressed or strengthened. This study makes three theoretical contributions to leader trait literature. First, we seek to enhance understanding of leader extraversion in team performance. The extant research on this topic [8, 11] has drawn on dominance complementarity theory to explain this relationship and has called for more research to investigate the motivational mechanisms linking leader extraversion and team performance. Our study, responding to this call, clarifies these mechanisms using distal-proximal motivation theory. Second, we examine the effects of situational characteristics (i.e., clarity) on the expression of leader extraversion. This second objective echoes the need for further research on how individual leader differences are integrated with situational characteristics [10, 12, 13]. Third, we explain goal clarity and process clarity and their effect on leadership, propose a new perspective on their mechanisms in the field of leadership, and enrich the existing research on goal clarity and process clarity. From the perspective of practice, this study also provides suggestions for the selection, promotion and training of team leaders.

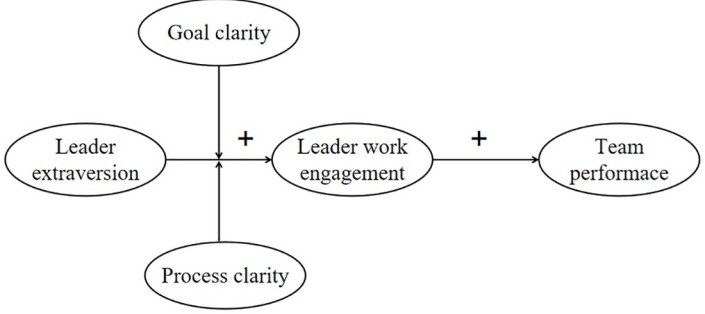

**Fig 1. Conceptual model.**

## Theoretical background and hypotheses

### Leader extraversion and team performance

Extraverted individuals are characterized as being active, talkative, sociable, assertive, energetic, cheerful, and socially confident [2, 5]. They are more likely to engage in activities that interact with and influence others in a dominant and assertive way. Extant research has established the positive relationships between leader extraversion and important leadership outcomes. For example, extraversion has been found to predict different domains of performance in managerial positions [14–16]. Several meta-analytic results also suggest significant and robust associations between extraversion as well as its lower-order traits and leadership effectiveness [1, 7, 17].

Beyond these established relationships, other underrepresented research attempts have been made to understand the associations between leader extraversion and team performance. Drawing on dominance complementarity theory, Grant, Gino, and Hofmann [8] find that the positive relationship between extraverted leadership and team performance is attenuated when employees are proactive, which identifies a condition in which the positive effect is reduced. A more recent study [11] draws on the same theoretical framework and examines the effects of leader extraversion and other two personality traits on team performance through team potency beliefs and through relationship identification with the leader. Their findings suggest that leader extraversion has a positive indirect effect on team in-role and extra-role performance via relational identification for high-power distance teams. Both studies shed light on certain situations when leader extraversion influences team performance. Accordingly,

Hypothesis 1: Leader extraversion has a positive relationship with team performance.

### The mediating effect of leader work engagement

According to distal-proximal motivational theory, motivation constructs should be ranked in a causal order based on their proximity to behavioral activities [18]. A personality characteristic is considered more distal than a motivation, since these characteristics affect a person's intention to engage in certain behaviors (e.g., motivation) [18]. This, then, seems to suggest that leadership attributes such as the Big Five traits indirectly influence leadership effectiveness through more proximal motivational mechanisms.

Work engagement is an affective-motivational work-related state characterized by vigor, dedication, and absorption [19, 20]. For employees, commitment to work as a motivating state can make the connection between personality and performance [9, 21, 22]. Extraverted individuals are more inclined to engage in social activities and experience positive emotions and are more able to invest themselves in their work, which translates into higher levels of performance [22]. Kanfer [18] identifies personality as a distal variable affecting performance, and individual motivational states as the explanatory mechanism between personality and performance. Thus, work engagement is a mediating variable linking extraverted personality and job performance.

The same mediation mechanism exists at the leadership for work engagement. Team performance is a reflection of the quality of a team leader's work, and team leader work engagement connects extraverted leadership with team performance. The reason this paper focuses on the leaders' own motivational state is that the impact of extraverted leadership on team performance is a study of the mechanisms of action of leadership personality, whereas most previous research has examined the mechanisms of action of leadership behavior. Leadership personality has a more distal impact on performance than leadership behavior and requires the state of the leader as a proximal variable to mediate [14, 18].

To integrate these associations, we drew on the distal-proximal motivational theory [9], which states that proximal individual differences act through motivational mechanisms explaining distal individual differences' effect on performance [9, 23]. Extraverted leaders elevate their work engagement by devoting and replenishing energy to leadership roles [24], which facilitates team performance by engaging in more positive leader activities. Accordingly,

Hypothesis 2: Leader work engagement mediates the positive relationship between leader extraversion and team performance.

## The moderating effect of goal clarity and process clarity

According to the situational strength theory [25, 26], the expression of individual differences and personality is constrained in strong situations where clear cues are given. Meyer et al. [27] define situational strength as "implicit or explicit cues provided by external entities regarding the desirability of potential behaviors" (p. 122). In strong situations, clear rules and cues in work environments provide uniform guidance for expected behaviors [26, 27], whereas in weak situations, work environments comprise unstructured social roles, decentralized organizational structures, and considerable discretion over work behaviors [10, 28]. Meyer et al. proposed four facets of situational strength: clarity, consistency, constraints, and consequences [27]. They defined clarity as the availability of cues about work-related obligations, consistency as the compatibility of work-related responsibilities and demands, constraints as the extent to which employees are allowed to determine when, where and how they work, and consequences as the degree to which decisions have clear positive or negative consequences for staff members [27, 29]. Situational strength determines how much a personality trait impacts behavior, with a strong situation overwhelming individual differences, and a weak situation having a greater impact [29].

It is important for individuals to know their own work goals and how to achieve them as they fulfill their task roles [26, 30]. Clear goals and processes help employees define their work goals and the steps they must take to accomplish them in terms of work characteristics, as well as their relationships with teammates and supervisors [30, 31]. Goal clarity and process clarity are in line with the characteristics of strong situations: job responsibilities are easier to understand, work goals and processes are compatible, individual decision-making is limited, and specific behaviors of employees contribute to better team performance [27, 32]. A strong work environment reduces personality differences among employees, encouraging individuals with their various characteristics to achieve work goals [33].

The leadership literature demonstrates the interactionist model, where situational parameters exert influence on the degree to which leader personality predicts leadership effectiveness [10, 13, 34]. For extraverted leaders, clarity of goals and process may not be a positive situation [10, 35]. Although clarity favors well-defined work environments, it may also prohibit leaders from expressing such personality traits as extraversion and the activation of motivation [10, 35]. In a situation where goals or processes are clearly defined, expectations of leader behaviors and outcomes are more likely to be uniform, so that leaders perform in similar ways in order to achieve both individual and team goals. However, when ambiguity is allowed in the work environment, leaders with a high level of extraversion are activated to take initiative and express greater autonomy. In this weak situation, extraverted leaders feel more comfortable and less constrained to express personality characteristics such as high levels of activity and energy. Their experience and expression of positive emotions is likely to lead them to devote more energy to leader roles and activities, and thus they have higher work engagement than their introverted counterparts. Accordingly,

Hypothesis 3a: Goal clarity moderates the positive relationship between leader trait extraversion and leader work engagement, such that the relationship is weaker when LGC is high rather than low.

Hypothesis 3b: Process clarity moderates the positive relationship between leader trait extraversion and leader work engagement, such that the relationship is weaker when LPC is high rather than low.

## Materials and methods

### Participants and procedures

All procedures were conducted in compliance with the American Psychology Association (APA) ethics code and approved by affiliations of the first and third authors (Beijing Institute of Petrochemical Technology), although the university did not have a formal Institutional Review Board. It should be noted that the study does not fall within the field of clinical psychology. In addition, all participants provided written informed consent when they were filling out the questionnaires.

This study used a questionnaire survey to collect data from eight companies in China, covering the information technology industry, the electric power industry, the financial industry, and the education industry. A researcher was invited by the eight companies to conduct leadership training for their management teams. We recruited the subjects in the training courses with the consent of the enterprises and on the basis of voluntary participation. Firstly, we obtained permission from the heads of the human resources departments and explained our purpose in collecting data to. Then we gave a verbal explanation of the research purpose to the respondents prior to data collection, and assured them that the data would remain confidential and would be used exclusively for this research. Finally, respondents had to read an informed consent document and check "I agree" before completing the questionnaire. They could withdraw at any time.

We conducted two rounds of questionnaire collection. During the training session, we distributed the link to complete the electronic questionnaire and collected the first round of data on-site, including demographics, occupational data, level of extraversion, and perceptions of goal clarity and process clarity. Four weeks later, the link to the second round of electronic questionnaires was distributed through the channels of the training courses' WeChat groups and the data was collected on the same day. The same group of leaders was asked to complete a second questionnaire on the subject of work engagement and perceived team performance.

The final sample of 226 valid respondents from 320 team leaders was predominantly male (72.6%) and had an average leadership tenure of 7.10 years (SD = 5.99). The age of respondents ranged from 23 to 56 years (M = 38.18, SD = 6.78). All possessed at least a high school diploma. In addition, respondents were diverse regarding team functions and industry: 26.1% research and development, 22.6% business support (including human resource, administration, legal, and financial controlling), 20.4% customer service, 16.4% production, and 14.6% marketing and sales; 32.3% information technology, 35.4% electricity and power, 23.5% finance, and 9.3% education.

### Measures

A group of six subject matter experts with psychology or human resource management background translated and back translated all measures from English to Simplified Chinese according to procedures from the International Test Commission (2017) [36]. All measures were self-report.

**Leader extraversion.**   Leader extraversion was measured by 10 items from Big-Five Factor Markers [37]. Response options ranged from 1 (*strongly disagree*) to 5 (*strongly agree*). An example item was"I feel comfortable around people."The alpha coefficient of the scale is .84.

**Goal clarity and process clarity.**   To measure goal clarity and process clarity, we adapted Sawyer et al.'s(1992) [31] goal and process clarity scale. A certainty scale (1 = *very uncertain*; 5 = *very certain*) was used to indicate the clarity of goals and process as perceived by leaders. An example item of goal clarity was"My responsibilities are clear", the alpha coefficient of the scale is .84. An example item of process clarity was"I know how to organize my daily tasks", the alpha coefficient of the scale is .87.

**Leader work engagement.**   We measured leader work engagement by Schaufeli et al.'s 9-item Utrecht Work Engagement Scale (UWES-9) [19]. Respondents indicated their extent of agreement on a scale ranging from 1 (*strongly disagree*) to 7 (*strongly agree*). An example item was"I devoted myself to my training and studies", the alpha coefficient of the scale is .88.

**Team performance.**   We used Van Der Vegt and Bunderson's 5-item team performance scale [38]. Leaders were asked to rate the performance of their teams by indicating their level of agreement on a scale ranging from 1 (*strongly disagree*) to 7 (*strongly agree*). An example item was"The performance of my team was excellent as a whole", the alpha coefficient of the scale is .96.

**Control variables.**   We controlled for demographic characteristics variables that may impact leader extraversion, leader work engagement, and team performance, including age, gender, education level and tenure with leader.

## Analyses

In this study, we used SPSS 26 and MPLUS 7.0 to analyze the 226 valid data. We conducted confirmatory factor analysis (CFA) using Mplus 7.0 to examine the discriminant validity of main variables. In order to test the common method bias, we conducted the Harman univariate analysis using SPSS 26 software. Then, we used hierarchical multiple regression to test the mediating effect and hierarchical moderated regression to test moderating, moderated mediating, and partial moderated mediating effects using SPSS 26. For all regression analyses, we entered age, gender, education, and leadership tenure as control variables. All continuous variables were centered to avoid multicollinearity. In addition, we used the PROCESS macro (Model 4; Hayes, 2017) and estimated 95% confidence intervals (CIs) based on 10,000 boot-strapped samples in order to assess the significance of mediating effects. Moreover, we estimated the conditional indirect effects based on the levels of LGC or LPC using Models 8 of PROCESS macro (Hayes, 2017, 2018) with 10,000 bootstrapped samples.

## Results

### Exploratory factor analysis

By using Mplus7, EFA was used to analyze the potential factor structure of the questionnaire. As shown in Table 1, Model fit indices were compared based on the theoretical study, and the optimal 5-factor model was selected. The factor loadings for the measured indexes were 0.919 at the maximum and 0.535 at the minimum, which met the criteria of 0.50~0.95.

### Confirmatory factor analysis

CFA was conducted on the survey items addressing five variables using the software Mplus 7.0. Using data from 226 questionnaires, we compared five alternative models with the baseline model, five factor Model 1. As shown in Table 2, CFA results indicated that the five-

**Table 1. Results for confirmatory factor analysis.**

| Model | $\chi^2$ | df | CFI | TLI | AIC | BIC | SRMR | RMSEA(90% CI) |
|---|---|---|---|---|---|---|---|---|
| Five-factor model | 762.53** | 401 | .90 | .87 | 16201.13 | 16981.01 | .04 | .06(.056,.007) |
| Four-factor model | 900.65** | 431 | .87 | .84 | 16328.45 | 17005.72 | .04 | .07(.063,.076) |
| Three-factor model | 1337.08** | 462 | .77 | .72 | 16755.91 | 17327.13 | .08 | .09(.086,.097) |
| Two-factor model | 1811.46** | 494 | .65 | .60 | 17249.79 | 17711.57 | .10 | .11(.103,.114) |
| One-factor model | 2621.75** | 527 | .44 | .41 | 18152.18 | 18501.07 | .13 | .13(.128,.138) |

*Note.*

** $p < .01$

**Table 2. Results for confirmatory factor analysis.**

| Model | $\chi^2$ | df | CFI | TLI | RMSEA | $\Delta\chi^2/\Delta df$ |
|---|---|---|---|---|---|---|
| Five-factor model | 920.95** | 507 | .91 | .90 | .06 | |
| Four-factor model 1[a] | 1620.85** | 521 | .76 | .74 | .10 | 49.99** |
| Four-factor model 2[b] | 1602.88** | 521 | .77 | .75 | .10 | 48.70** |
| Three-factor model[c] | 1697.13** | 524 | .75 | .73 | .10 | 31.42** |
| Two-factor model[d] | 2149.89** | 526 | .65 | .62 | .12 | 226.38** |
| One-factor model[e] | 3049.79** | 527 | .45 | .42 | .15 | 899.90** |

*Note.*

** $p < .01$

[a] This model combines, from the five-factor model, leader extraversion and process clarity to form one factor.

[b] This model combines, from the five-factor model, leader extraversion and goal clarity to form one factor.

[c] This model combines leader extraversion, goal clarity, and process clarity into one factor.

[d] This model combines leader extraversion, goal clarity, process clarity, and leader work engagement into one factor.

[e] This model combines all variables into one factor.

factor model consisting of leader extraversion, goal clarity, process clarity, leader work engagement, and team performance fit the data better than alternative models.

## Common method bias testing

Harman's single-factor test was conducted to confirm the possibility of CMB. According to Podsakoff et al. (2003) [39], single-factor cumulative variance explanation rates of less than 40% are acceptable. We adopted SPSS 26 to perform Harman's single-factor analysis, the results showed that the cumulative variance explanation rate of the first precipitation factor was 30.7% (< 40%), indicating that the common method bias was acceptable.

## Descriptive statistics

Table 3 shows descriptive statistics and correlations among study variables. As expected, leader extraversion was positively associated with team performance (r = .25, $p < .01$), which supported Hypothesis 1.

## Hypothesis testing

Table 4 presents results for the mediation effect of leader work engagement. Leader extraversion was significantly associated with leader work engagement (M1b; ß = .32, p < .01) and was a significant predictor of team performance (M2b; ß = .25, p < .01). Leader work engagement

**Table 3. Means, standard deviations, and correlations.**

| Variables | M | SD | 1 | 2 | 3 | 4 | 5 | 6 | 7 | 8 | 9 |
|---|---|---|---|---|---|---|---|---|---|---|---|
| 1. Gender | 0.27 | 0.45 | | | | | | | | | |
| 2. Age | 38.18 | 6.78 | -.03 | | | | | | | | |
| 3. Education | 2.82 | 0.79 | .16* | -.45** | | | | | | | |
| 4. Leadership tenure | 7.10 | 5.99 | -.10 | .63** | -.23** | | | | | | |
| 5. Leader extraversion | 3.49 | 0.54 | -.03 | .05 | .04 | .04 | *.84* | | | | |
| 6. Goal clarity | 4.12 | 0.50 | .12 | -.02 | .10 | -.02 | .32** | *.84* | | | |
| 7. Process clarity | 3.99 | 0.52 | .17* | .06 | .13 | .10 | .38** | .70** | *.87* | | |
| 8. Leader work engagement | 4.83 | 0.87 | .04 | .01 | .14* | .04 | .33** | .34** | .47** | *.88* | |
| 9. Team performance | 5.25 | 1.13 | .14* | .17** | -.07 | .14* | .25** | .40** | .46** | .42** | *.96* |

*Note*. N = 226. Numbers in the lower diagonal are correlations. Cronbach's alpha estimates are reported on the diagonal in bold and italic. Gender: 0 = male; 1 = female. Education: 1 = *high school and below*, 2 = *associate degree*, 3 = *bachelor's degree*, 4 = *master's degree and above*. M = mean, SD = standard deviation.

\* $p < .05$.

\*\* $p < .01$.

was also a significant predictor of team performance (M2c; ß = .42, p < .01). When team performance was regressed on both leader extraversion and leader work engagement, with age, gender, education, and leadership tenure controlled for, only leader work engagement was a significant predictor (M2d; ß = .38, p < .01). Further bootstrapping analyses revealed positive and significant indirect effect of leader extraversion on team performance through leader work engagement (indirect effect = .26, SE = .08, 95% CI [.13, .46]). Therefore, Hypothesis 2 was supported.

Table 5 shows results of the moderating effects. Results suggested that the interaction of leader extraversion and goal clarity had significant and negative effect on leader work engagement (M3c; ß = -.20, p < .01) with main effects controlled for, demonstrating that goal clarity moderated the positive relationship between leader extraversion and leader work engagement. Simple slope analyses indicated that the effect of leader extraversion on leader work engagement was stronger when goal clarity was low (simple slope = .49, p < .01) rather than high (simple slope = .21, ns). Similarly, results showed that process clarity did moderate the positive

**Table 4. Mediating effect of leader work engagement.**

| Variables | Leader work engagement | | Team performance | | | |
|---|---|---|---|---|---|---|
| | M1a | M1b | M2a | M2b | M2c | M2d |
| Age | .07 | .05 | .12 | .10 | .09 | .08 |
| Sex | .02 | .03 | .15* | .16* | .15* | .15* |
| Education | .18* | .15* | -.02 | -.05 | -.10 | -.10 |
| Tenure with leader | .03 | .03 | .07 | .07 | .06 | .06 |
| Leader extraversion | | .32** | | .25** | | .12 |
| Leader work engagement | | | | | .42** | .38** |
| $R^2$ | .01 | .11** | .04* | .10** | .21** | .22** |
| $\Delta R^2$ | | .10** | | .06** | .11** | .12** |

*Note*. N = 226. All regression coefficients reported in this table are standardized ($\beta$).

\* $p < .05$.

\*\* $p < .01$.

**Table 5. Moderating effect of goal clarity and process clarity.**

| Variables | M3a | M3b | M3c | M3d |
|---|---|---|---|---|
| *Control variables* | | | | |
| Age | .07 | .05 | .01 | .00 |
| Sex | .02 | .03 | .01 | -.03 |
| Education | .17* | .15* | .10 | .07 |
| Tenure with leader | .03 | .03 | .04 | .00 |
| *Main items* | | | | |
| Leader extraversion | | .32** | .21** | .15* |
| Goal clarity | | | .18* | |
| Process clarity | | | | .34** |
| *Interaction items* | | | | |
| Leader Extraversion * Goal clarity | | | -.20** | |
| Leader Extraversion * Process clarity | | | | -.15* |
| $R^2$ | .01 | .11** | .21** | .27** |
| $\Delta R^2$ | | .10** | .10** | .16** |

*Note.* N = 226. All regression coefficients reported in this table are standardized ($\beta$). Dependent variable: leader work engagement.

* $p < .05$.

** $p < .01$.

relationship between leader extraversion and leader work engagement (M3d; ß = -.15, p < .01), such that the effect of leader extraversion on leader work engagement was strengthened when the level of process clarity was low (simple slope = .36, p < .01) rather than high (simple slope = .12, ns). Figs 2 and 3 show the interaction plots, which are consistent with the findings. Thus, Hypotheses 3a and 3b were supported as well.

To test moderated mediation, we estimated the conditional indirect effect of leader extraversion on team performance through leader work engagement at high and low values (±1 *SD*) of goal clarity and process clarity in two separate models. Results suggested that the indirect effect of leader extraversion on team performance via leader work engagement was significant and positive when goal clarity was low (indirect effect = .25, SE = .08, 95% CI [.13, .41]), but not significant when goal clarity was high (indirect effect = .11, SE = .07, 95% CI [-.01, .27]). Similarly, the indirect effect of leader extraversion through leader work engagement was positive and significant when process clarity was low (indirect effect = .18, SE = .07, 95% CI [.06, .31]), but not significant process clarity was high (indirect effect = .06, SE = .07, 95% CI [-.06, .21]). In addition, CIs for indices of moderated mediation indicated that the mediating effect of leader work engagement was moderated, respectively, by goal clarity (difference = -.14, SE = .09, 95% CI [-.36, -.03] and process clarity (difference = -.17, SE = .08, 95% CI [-.31, -.01]).

## Discussion

### Theoretical contributions

This study's primary theoretical contribution is providing an empirical explanation of how leader extraversion influences team performance by acting as an actionable motivational mechanism for leader work engagement, enriching previous research on the mechanisms of extraverted leadership influence on team performance. Scholars have called for research that explores the influence processes of leadership traits [1, 4]. The mechanisms of influence on

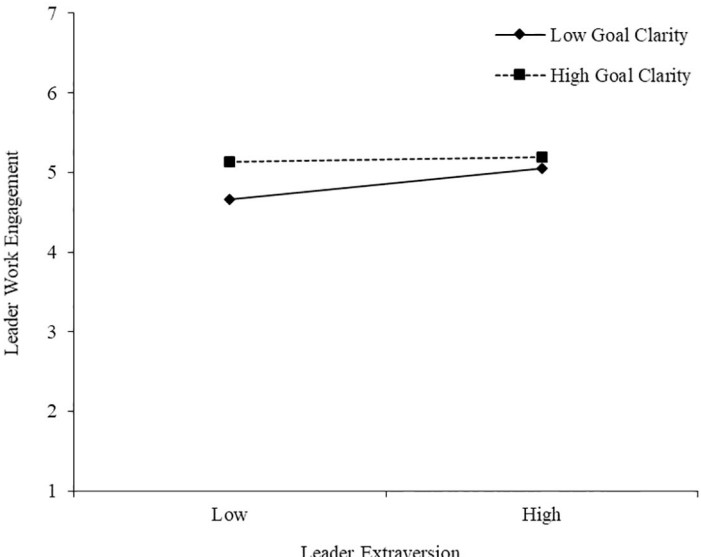

**Fig 2. Moderating effect of goal clarity on the relationship between leader extraversion and leader work engagement.**

extraverted leadership-the type of leadership trait that has received the most attention [1]-are not clear. In this study, following Kanfer's [18] model of motivational states linking personality and work outcomes, work engagement as a motivational variable was found to be a mediating variable between extraversion and team performance. In particular, we look at leader work engagement to further enrich the study of team performance antecedent variables and leadership with the perspective of leaders' motivational states.

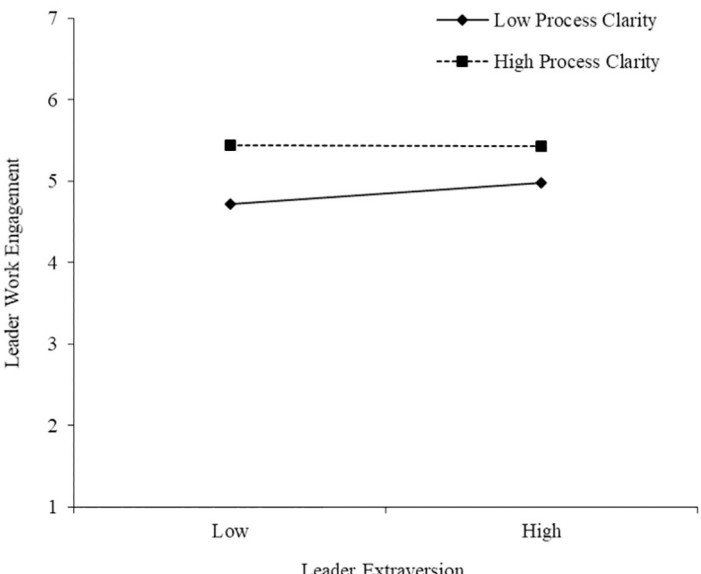

**Fig 3. Moderating effect of process clarity on the relationship between leader extraversion and leader work engagement.**

The second key contribution of this study is the finding that goal clarity and process clarity limit the role of extraverted leader personality, providing empirical evidence for the validity of situational strength theory. Situational strength theory suggests that strong situations can limit the role of personality in predicting individual behavior and outcomes [27], and Judge et al. [10] suggest that applying this theory to discussion of the limitations of situations on the role of personality requires a clear understanding of what the situation is and how it affects personality. Roberts et al. [40] point out that it is important to find situations that are truly relevant to the role of personality. To this end, this paper looks at four aspects of the concept of situational strength and identifies goal clarity and process clarity as important factors in the work environment. Further, these findings are combined with a consideration of characteristics of the extraverted personality, which requires dominance and enjoys challenging and changing environments, in order to conclude that goal clarity and process clarity are the right contexts to influence the functioning of the extraversion. The findings of this paper provide empirical evidence for the theory of situational strengths and offer implications for future research.

The third theoretical contribution of this study is the extension of the boundaries of the influence mechanisms of extraverted leadership. Current research on leader extraversion has been more concerned with extraversion as a contextual moderator of the relationship between leadership style and job outcomes [41, 42], overlooking the contexts that influence the expression of extraverted leadership traits [8, 34]. We introduce goal clarity and process clarity of leadership as important contexts influencing the role of extraverted leadership on performance and find that the positive effect of extraversion on team performance is weakened in situations where goal clarity and process clarity are high. The findings of this paper question the assertion that extraverted leaders are always effective and provide empirical evidence for that the characteristics of extraverted leadership vary depending on the situation.

## Practical contributions

Our findings suggest important practical implications for the organizations seeking to identify effective team leaders. In the current era of volatility, unpredictability, complexity, and ambiguity, leaders need to navigate teams to overcome challenges and achieve various business goals. The performance advantage of extraversion in managerial and leadership positions has been claimed and proven already [43, 44]. In the present study, the advantage of leader extraversion on work engagement and team performance is seen to stand out when levels of goal clarity or process clarity are low. Given the importance of leader extraversion on leader effectiveness and team performance, it is crucial for organizations to identify team leaders with moderate or high levels of extraversion.

In addition, given that extraverted leaders like to be challenged by change, it is important to note that the ability of extraverted leaders to function may change at different stages of business development. For example, in the start-up phase, an extraverted leader can make full use of his or her energy in a changing environment, while in the maturing phase, the extraverted leader's energy may be limited by the rules. Therefore, team leaders with different levels of extroversion need to be selected at different stages of business development.

Finally, this paper offers new suggestions for leadership improvement. Favoring an extraverted, transformational approach to leadership is a crucial goal for developing effective leaders. And as the conclusions of this article demonstrate, leadership power change theory and situational leadership theory are both valid. When upper management assigns tasks to extraverted leaders, they should make full use of the strengths of extraverted leaders by giving, them more freedom to make their own arrangements.

### Limitations and future research

The present study has its limitations. First, the measure of the dependent variable in this study is self-reported, which is more likely to suffer from subjectivity. Future research that uses additional measures for team performance (e.g. the entire team evaluated for team performance), and more objective measures (e.g. sales revenue for sales team) where possible would add both theoretical and practical significance to the current findings. Second, we measured extraversion as a unified dimension in this study and did not consider subdimensions of extraversion when collecting data. We suggest that future research could investigate lower-order traits [7] and different aspects [17] of extraversion for more insights. Third, although our data comprised occupationally heterogenous team leaders across several industries, we cannot guarantee the generalization of findings to all occupations or all other cultures. In the future, the sample could be expanded to improve the external validity of the findings. For example, this study was conducted in a Chinese cultural context, which emphasizes modesty and understatement, and thus the data may reflect cultural differences in the impact of extraverted leadership on team performance. Future research could investigate this topic in a more culturally diverse context and could also attempt to explore the role of different cultural values in the relationship between leadership extroversion and team performance.

## Conclusion

We propose and test a moderated mediation model that explains the relationship between leader extraversion and team performance. We find that leader work engagement mediate the positive effect of leader extraversion on team performance. Moreover, the relationship between leader extraversion and leader work engagement is stronger when the level of goal clarity or process clarity is low than high. We also empirically validate this model such that low level of goal clarity or process clarity strengthen the positive effect of leader extraversion on team performance through leader work engagement. Future research expanding the understandings of leader extraversion (and/or other personality traits) and team performance is warranted.

## Supporting information

**S1 File. Data set of this study.**
(SAV)

## Acknowledgments

We thank Yang Yang for helpful comments and suggestions for revisions on an early version of this article.

## Author Contributions

**Conceptualization:** Jun Zhang, Kui Yin.

**Data curation:** Jun Zhang.

**Formal analysis:** Jun Zhang.

**Funding acquisition:** Jun Zhang.

**Investigation:** Jun Zhang, SiQi Li.

**Methodology:** Kui Yin.

**Writing – original draft:** Jun Zhang, SiQi Li.

**Writing – review & editing:** Jun Zhang, Kui Yin.

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
