## [Decision Letter · Decision Letter 0]

28 Aug 2022

PONE-D-22-19951Leader Extraversion and Team Performance: A Moderated Mediation ModelPLOS ONE

Dear Dr. Zhang,

Thank you for submitting your manuscript to PLOS ONE. After careful consideration, we feel that it has merit but does not fully meet PLOS ONE’s publication criteria as it currently stands. Therefore, we invite you to submit a revised version of the manuscript that addresses the points raised during the review process.

Reviewer 1:

Although the paper is well written but requires several changes, a few of them are as follows

**Abstract**

Add a background sentence before stating the objective of the study. In addition, when and how data was collected [I mean how respondents were selected; in which month or year data was collected]. Besides, also highlight which software has been used for the analysis. Finally, add a conclusion sentence and add some implications.

**Introduction**

This section was very limited. There is a need to add more content [why this research is needed] and highlight the research gap by covering each relationship proposed in the model. Based on the research gap, draw the research questions and objective and add the contribution paragraph in the last. I will suggest you cite recent work instead of old work. I will suggest moving to figure 1 in the next section's theoretical background after the hypotheses end.

Theoretical background and hypotheses

I could not find any explanation for the distal-proximal motivational and situational strength theories. There is a need to define the assumptions of both theories first. After that, based on theoretical argument, construct your hypotheses. Referring to line 68 [author argues no work]. However, several studies investigate the mediation of work engagement.

Tanskanen, J., Mäkelä, L. & Viitala, R. Linking Managerial Coaching and Leader–Member Exchange on Work Engagement and Performance. *J Happiness Stud* **20**, 1217–1240 (2019). https://doi.org/10.1007/s10902-018-9996-9

Gutermann, D., Lehmann-Willenbrock, N., Boer, D., Born, M. and Voelpel, S.C. (2017), How Leaders Affect Followers’ Work Engagement and Performance: Integrating Leader−Member Exchange and Crossover Theory. Brit J Manage, 28: 299-314. https://doi.org/10.1111/1467-8551.12214

**Methodology**

Add a sampling section and explain further how respondents have been selected. In addition, add each constructs item used in the study.

**Results **

Divide the results section into the measurement model and structural model. In particular, in the measurement model, use EFA to better look into the multicollinearity issues; the current results are insufficient to comment further.

**Discussion **

This section needs substantial changes. Initially, divide this section into three sub-parts: theoretical and practical implications, limitations, and future research. Secondly, discuss your result in the context of prior studies which is missing. Theoretical and practical implications require to be supported by citations.  

Reviewer 2:

Changes which must be made before publication:

I found the paper very interesting. But I have some comments which might help you improve your work:

1. I found sentence structure problems in your text that must be corrected.

In introduction part:

2. the arguments of your introduction do not arrive at your research questions.

3. There are various studies on leader characteristics and team performance how your study is different from them?

4. Proper attention must be given to gap identification, problem, and research questions in your study

5. Research gap for mediation and moderation role must also be discussed.

6. Conceptual model of the study must be discussed in terms of research gap in existing studies and why it is important.

In materials and methods Part:

7. Please provide the justification for using this measure of dependent variable - self-report. What are other available ways and why this one is better measure?

8. In discussion part, the discussion of findings is poorly written. It would be better if you discuss the novelty of the findings and its relevant insights with divergent views that may differ from other studies but helpful to your readers.

9. Also update the discussion part with recommendations and insights from recent literature.

10. The conclusion of the study is very shallow and merely repeating results. What are the insights of this study and what are its practical implications to real world.

11. How can you enhance the generalizability of your study? Kindly work on this.

We look forward to receiving your revised manuscript.

Kind regards,

Mingyue Fan, Ph.D.

Academic Editor

PLOS ONE

Journal Requirements:

Reviewers' comments:

Reviewer's Responses to Questions

**Comments to the Author**

1. Is the manuscript technically sound, and do the data support the conclusions?

Reviewer #1: Partly

Reviewer #2: Partly

2. Has the statistical analysis been performed appropriately and rigorously? 

Reviewer #1: No

Reviewer #2: Yes

3. Have the authors made all data underlying the findings in their manuscript fully available?

Reviewer #1: No

Reviewer #2: Yes

4. Is the manuscript presented in an intelligible fashion and written in standard English?

Reviewer #1: No

Reviewer #2: Yes

5. Review Comments to the Author

Reviewer #1: Although the paper is well written but requires several changes, a few of them are as follows

Abstract

Add a background sentence before stating the objective of the study. In addition, when and how data was collected [I mean how respondents were selected; in which month or year data was collected]. Besides, also highlight which software has been used for the analysis. Finally, add a conclusion sentence and add some implications.

Introduction

This section was very limited. There is a need to add more content [why this research is needed] and highlight the research gap by covering each relationship proposed in the model. Based on the research gap, draw the research questions and objective and add the contribution paragraph in the last. I will suggest you cite recent work instead of old work. I will suggest moving to figure 1 in the next section's theoretical background after the hypotheses end.

Theoretical background and hypotheses

I could not find any explanation for the distal-proximal motivational and situational strength theories. There is a need to define the assumptions of both theories first. After that, based on theoretical argument, construct your hypotheses. Referring to line 68 [author argues no work]. However, several studies investigate the mediation of work engagement.

Tanskanen, J., Mäkelä, L. & Viitala, R. Linking Managerial Coaching and Leader–Member Exchange on Work Engagement and Performance. J Happiness Stud 20, 1217–1240 (2019). https://doi.org/10.1007/s10902-018-9996-9

Gutermann, D., Lehmann-Willenbrock, N., Boer, D., Born, M. and Voelpel, S.C. (2017), How Leaders Affect Followers’ Work Engagement and Performance: Integrating Leader−Member Exchange and Crossover Theory. Brit J Manage, 28: 299-314. https://doi.org/10.1111/1467-8551.12214

Methodology

Add a sampling section and explain further how respondents have been selected. In addition, add each constructs item used in the study.

Results

Divide the results section into the measurement model and structural model. In particular, in the measurement model, use EFA to better look into the multicollinearity issues; the current results are insufficient to comment further.

Discussion

This section needs substantial changes. Initially, divide this section into three sub-parts: theoretical and practical implications, limitations, and future research. Secondly, discuss your result in the context of prior studies which is missing. Theoretical and practical implications require to be supported by citations.

Reviewer #2: Changes which must be made before publication:

I found the paper very interesting. But I have some comments which might help you improve your work:

1. I found sentence structure problems in your text that must be corrected.

In introduction part:

2. the arguments of your introduction do not arrive at your research questions.

3. There are various studies on leader characteristics and team performance how your study is different from them?

4. Proper attention must be given to gap identification, problem, and research questions in your study

5. Research gap for mediation and moderation role must also be discussed.

6. Conceptual model of the study must be discussed in terms of research gap in existing studies and why it is important.

In materials and methods Part:

7. Please provide the justification for using this measure of dependent variable - self-report. What are other available ways and why this one is better measure?

8. In discussion part, the discussion of findings is poorly written. It would be better if you discuss the novelty of the findings and its relevant insights with divergent views that may differ from other studies but helpful to your readers.

9. Also update the discussion part with recommendations and insights from recent literature.

10. The conclusion of the study is very shallow and merely repeating results. What are the insights of this study and what are its practical implications to real world.

11. How can you enhance the generalizability of your study? Kindly work on this.

6. PLOS authors have the option to publish the peer review history of their article (what does this mean?). If published, this will include your full peer review and any attached files.

Reviewer #1: **Yes: **Sikandar Ali Qalati

Reviewer #2: No

---

## [Author Response · Author response to Decision Letter 0]

10 Oct 2022

Respected Editor and Reviewers, 

Thanks for the editors and reviewers’comments and suggestions. In line with the reviewers’ comments, we accepted most of the editors’ suggestion and carefully revised them. 

We appreciate the Editors/Reviewers‟ earnest work, and hope that our changes meet approval. Once again, thank you very much for your comments and suggestions. 

The main corrections in the paper and the responds to the reviewers’ comments are 

listed below. 

Kind regards, 

Auhtors

Reviewer 1:

Abstract

Add a background sentence before stating the objective of the study. In addition, when and how data was collected [I mean how respondents were selected; in which month or year data was collected]. Besides, also highlight which software has been used for the analysis. Finally, add a conclusion sentence and add some implications.

Response to the Reviewer 1 about Abstract:

Many thanks to the reviewers for their comments. In accordance with your comments, we have added relevant information to the abstract. The details are as follows:“Extraversion is the best and most consistent predictor of important leadership outcomes. However,there has been little exploration and examination of the mechanisms underlying the effects of extraverted leadership on performance. Drawing on distal-proximal motivational theory and situational strength theory, the present study proposes and examines a moderated mediation model that explains how leader extraversion affects team performance and how situational characteristics strengthen or constrain this relationship. Respondents were recruited through management team training courses run by the eight Chinese companies. We conducted two rounds of electronic questionnaire collection. The first round of data was collected during the training session. Four weeks later, we collected the data through the training courses’ WeChat groups. Data collected from 226 Chinese team leaders was analyzed using SPSS 26 and Mplus 7. We find that leader extraversion predicts team performance through a motivational mechanism operationalized as leader work engagement. We further find that goal clarity and process clarity play an important role in strengthening the positive effect of leader extraversion on leader work engagement as well as the motivational mechanism, providing an empirical explanation of how leader extraversion affects team performance through a motivational mechanism operationalized as leader work engagement. We also explore how two potential situational characteristics, operationalized as goal clarity and process clarity of leaders, affect the relationship between leader extraversion and leader work engagement as well as the motivational mechanism. Addionally,the findings suggest important practical implications for the organizations seeking to identify effective team leaders. ”

Introduction

This section was very limited. There is a need to add more content [why this research is needed] and highlight the research gap by covering each relationship proposed in the model. Based on the research gap, draw the research questions and objective and add the contribution paragraph in the last. I will suggest you cite recent work instead of old work. I will suggest moving to figure 1 in the next section's theoretical background after the hypotheses end.

Response to the Reviewer 1 about Introduction:

We thank the reviewer for constructive suggestions and for providing us with clear directions. According to the reviewer's revision suggestions, the authors have revised four areas: (1) enhanced discourse on research gaps ; (2) reorganised contributions and added in the last ; (3)cited recent literature; and (4) moved Figure 1 to the end of the hypothesis. The details are as follows:

“Extraverted personality has long been acclaimed as the most effective personality variable for predicting leadership[1]. Within organizations, leaders influence others through interactions to motivate them to reach team and organizational goals. This requires leaders to possess characteristics consistent with extraversion: sociablility, contagiours energy, assertiveness, and gregariousness [2-5 ]. Two seminal meta-analytic research papers[1,6] have established that of the five-factor model (FFM) of personality traits, extraversion is the best and most consistent predictor of important leadership outcomes (i.e., leader emergence, leadership effectiveness, and transformational leadership). Furthermore, one most recent meta-analysis [7] focusing solely on extraversion provides more detailed and in-depth findings showing that extraversion and its lower-order traits are consistent predictors of leader emergence, leader behaviors (for both transformational and transactional leadership), and leadership effectiveness. Compared to other traits, thus, extraverted personality has an advantage in predicting leadership and performance.

Despite its obvious advantages for working environments, there has been limited research to date on the mechanisms underlying the effects of extraverted leadership on performance [8]. Research on how extraverted leadership affects team performance and the circumstances in which the effects of extraverted leadership are inhibited or enhanced remains limited. Judge et al [1] called for further research to focus on processes of influence of leadership traits and contexts in which leadership traits have an effect. In particular, the mechanisms of influence of the Big Five personality traits are directly related to the entire field of leadership traits research [1,4]. This article, therefore, explores the process of influencing team performance and its contextual variables in terms of extraverted personality.

In this article, we draw on earlier work on the associations between leader extraversion and team performance and proposed our hypotheses from a theoretical perspective that was different from those prior works. Drawing on the distal-proximal motivation theory[9] and the situational strength theory [10], we develop a moderated mediation model (Fig 1) that helps explain both how leader extraversion affects team performance and when such effects are suppressed or strengthened. This study makes three theoretical contributions to leader trait literature. First, we seek to enhance understanding of leader extraversion in team performance. The extant research on this topic[8.11] has drawn on dominance complementarity theory to explain this relationship and has called for more research to investigate the motivational mechanisms linking leader extraversion and team performance. Our study, responding to this call,clarifies these mechanisms using distal-proximal motivation theory. Second,we examine the effects of situational characteristics (i.e., clarity) on the expression of leader extraversion. This second objective echoes the need for further research on how individual leader differences are integrated with situational characteristics[10,12-13] .Third, we explain goal clarity and process clarity and their effect on leadership,propose a new perspective on their mechanisms in the field of leadership,and enrich the existing research on goal clarity and process clarity. From the perspective of practice, this study also provides suggestions for the selection, promotion and training of team leaders.  ”

Theoretical background and hypotheses

I could not find any explanation for the distal-proximal motivational and situational strength theories. There is a need to define the assumptions of both theories first. After that, based on theoretical argument, construct your hypotheses. Referring to line 68 [author argues no work]. However, several studies investigate the mediation of work engagement.

Response to the Reviewer 1 about Theoretical background and hypotheses:

Many thanks to the reviewers for suggestions. As you pointed out, we didn’t provide a very clear explanation of the theoretical underpinnings of this section, and the statement in line 68 was also inaccurate.The theoretical statements have been added and this section has been rewritten.

The Mediating Effect of Leader Work Engagement

“According to distal-proximal motivational theory, motivation constructs should be ranked in a causal order based on their proximity to behavioral activities[18]. A personality characteristic is considered more distal than a motivation, since these characteristics affect a person's intention to engage in certain behaviors (e.g., motivation) [18]. This, then, seems to suggest that leadership attributes such as the Big Five traits indirectly influence leadership effectiveness through more proximal motivational mechanisms.

Work engagement is an affective-motivational work-related state characterized by vigor, dedication, and absorption[19-20]. For employees, commitment to work as a motivating state can make the connection between personality and performance[9,21-22].Extraverted individuals are more inclined to engage in social activities and experience positive emotions and are more able to invest themselves in their work, which translates into higher levels of performance [22]. Kanfer [18] identifies personality as a distal variable affecting performance, and individual motivational states as the explanatory mechanism between personality and performance. Thus, work engagement is a mediating variable linking extraverted personality and job performance.

The same mediation mechanism exists at the leadership for work engagement. Team performance is a reflection of the quality of a team leader's work, and team leader work engagement connects extraverted leadership with team performance. The reason this paper focuses on the leaders’ own motivational state is that the impact of extraverted leadership on team performance is a study of the mechanisms of action of leadership personality, whereas most previous research has examined the mechanisms of action of leadership behavior. Leadership personality has a more distal impact on performance than leadership behavior and requires the state of the leader as a proximal variable to mediate[14,18].”

The Moderating Effect of Goal Clarity and Process Clarity 

“According to the situational strength theory[25-26], the expression of individual differences and personality is constrained in strong situations where clear cues are given. Meyer et al. [27]define situational strength as “implicit or explicit cues provided by external entities regarding the desirability of potential behaviors” (p. 122). In strong situations, clear rules and cues in work environments provide uniform guidance for expected behaviors [26-27], whereas in weak situations, work environments comprise unstructured social roles, decentralized organizational structures, and considerable discretion over work behaviors[10,28].） Meyer et al. proposed four facets of situational strength: clarity, consistency, constraints, and consequences[27]. They defined clarity as the availability of cues about work-related obligations, consistency as the compatibility of work-related responsibilities and demands, constraints as the extent to which employees are allowed to determine when, where and how they work, and consequences as the degree to which decisions have clear positive or negative consequences for staff members[27,29]. Situational strength determines how much a personality trait impacts behavior, with a strong situation overwhelming individual differences, and a weak situation having a greater impact[29].

It is important for individuals to know their own work goals and how to achieve them as they fulfill their task roles[26,30].Clear goals and processes help employees define their work goals and the steps they must take to accomplish them in terms of work characteristics, as well as their relationships with teammates and supervisors[30-31]. Goal clarity and process clarity are in line with the characteristics of strong situations: job responsibilities are easier to understand, work goals and processes are compatible, individual decision-making is limited, and specific behaviors of employees contribute to better team performance[27,32]. A strong work environment reduces personality differences among employees, encouraging individuals with their various characteristics to achieve work goals[33].

The leadership literature demonstrates the interactionist model, where situational parameters exert influence on the degree to which leader personality predicts leadership effectiveness[10,13,34]. For extraverted leaders, clarity of goals and process may not be a positive situation[10,35]. Although clarity favors well-defined work environments, it may also prohibit leaders from expressing such personality traits as extraversion and the activation of motivation[10,35]. In a situation where goals or processes are clearly defined, expectations of leader behaviors and outcomes are more likely to be uniform, so that leaders perform in similar ways in order to achieve both individual and team goals. However, when ambiguity is allowed in the work environment, leaders with a high level of extraversion are activated to take initiative and express greater autonomy. In this weak situation, extraverted leaders feel more comfortable and less constrained to express personality characteristics such as high levels of activity and energy. Their experience and expression of positive emotions is likely to lead them to devote more energy to leader roles and activities, and thus they have higher work engagement than their introverted counterparts.”

Methodology

Add a sampling section and explain further how respondents have been selected. In addition, add each constructs item used in the study.

Response to the Reviewer 1 about Methodology:

Many thanks for the suggestions! Your comments help us to refine the data collection process and enhance the rigor of the study.We revised the paper by adding relevant information.

“This study used a questionnaire survey to collect data from eight companies in China, covering the information technology industry, the electric power industry, the financial industry, and the education industry. A researcher was invited by the eight companies to conduct leadership training for their management teams. We recruited the subjects in the training courses with the consent of the enterprises and on the basis of voluntary participation. Firstly, we obtained permission from the heads of the human resources departments and explained our purpose in collecting data to. Then we gave a verbal explanation of the research purpose to the respondents prior to data collection, and assured them that the data would remain confidential and would be used exclusively for this research.Finally, respondents had to read an informed consent document and check "I agree" before completing the questionnaire. They could withdraw at any time.

We conducted two rounds of questionnaire collection. During the training session, we distributed the link to complete the electronic questionnaire and collected the first round of data on-site, including demographics, occupational data, level of extraversion, and perceptions of goal clarity and process clarity. Four weeks later, the link to the second round of electronic questionnaires was distributed through the channels of the training courses’ WeChat groups and the data was collected on the same day. The same group of leaders was asked to complete a second questionnaire on the subject of work engagement and perceived team performance.”

“Leader extraversion. Leader extraversion was measured by 10 items from Big-Five Factor Markers. Response options ranged from 1 (strongly disagree) to 5 (strongly agree). An example item was“I feel comfortable around people.”The alpha coefficient of the scale is .84.

Goal clarity and process clarity. To measure goal clarity and process clarity, we adapted Sawyer et al.’s(1992) goal and process clarity scale. A certainty scale (1 = very uncertain; 5 = very certain) was used to indicate the clarity of goals and process as perceived by leaders. An example item of goal clarity was“My responsibilities are clear”, the alpha coefficient of the scale is .84. An example item of process clarity was“I know how to organize my daily tasks”, the alpha coefficient of the scale is .87. 

Leader work engagement. We measured leader work engagement by Schaufeli et al.’s 9-item Utrecht Work Engagement Scale (UWES-9). Respondents indicated their extent of agreement on a scale ranging from 1 (strongly disagree) to 7 (strongly agree). An example item was“I devoted myself to my training and studies”, the alpha coefficient of the scale is .88. 

Team performance. We used Van Der Vegt and Bunderson’s 5-item team performance scale. Leaders were asked to rate the performance of their teams by indicating their level of agreement on a scale ranging from 1 (strongly disagree) to 7 (strongly agree). An example item was“The performance of my team was excellent as a whole”, the alpha coefficient of the scale is .96. 

Control variables. We controlled for demographic characteristics variables that may impact leader extraversion, leader work engagement, and team performance, including age, gender, education level and tenure with leader.”

Results

Divide the results section into the measurement model and structural model. In particular, in the measurement model, use EFA to better look into the multicollinearity issues; the current results are insufficient to comment further.

Response to the Reviewer 1 about Results:

We thank the reviewer for constructive suggestions and for providing us with clear directions. According to the reviewer's revision suggestions, the authors have made the corresponding changes: (1) supplemented the reliability test results in the measurement section ; (2) added the EFA test in the results section. Further details can be found in the red-labeled section in the text.

Discussion

This section needs substantial changes. Initially, divide this section into three sub-parts: theoretical and practical implications, limitations, and future research. Secondly, discuss your result in the context of prior studies which is missing. Theoretical and practical implications require to be supported by citations.  

Response to the Reviewer 1 about Discussion:

Many thanks for the suggestions! As you pointed out, our discussion in this section is not deep and sufficient. We have rewritten this section with the following details.

“Theoretical contributions 

This study’s primary theoretical contribution is providing an empirical explanation of how leader extraversion influences team performance by acting as an actionable motivational mechanism for leader work engagement, enriching previous research on the mechanisms of extraverted leadership influence on team performance. Scholars have called for research that explores the influence processes of leadership traits [1,4].The mechanisms of influence on extraverted leadership-the type of leadership trait that has received the most attention [1]-are not clear. In this study, following Kanfer's [18] model of motivational states linking personality and work outcomes, work engagement as a motivational variable was found to be a mediating variable between extraversion and team performance. In particular, we look at leader work engagement to further enrich the study of team performance antecedent variables and leadership with the perspective of leaders' motivational states.

The second key contribution of this study is the finding that goal clarity and process clarity limit the role of extraverted leader personality, providing empirical evidence for the validity of situational strength theory. Situational strength theory suggests that strong situations can limit the role of personality in predicting individual behavior and outcomes [27], and Judge et al [10] suggest that applying this theory to discussion of the limitations of situations on the role of personality requires a clear understanding of what the situation is and how it affects personality. Roberts et al. [40] point out that it is important to find situations that are truly relevant to the role of personality. To this end, this paper looks at four aspects of the concept of situational strength and identifies goal clarity and process clarity as important factors in the work environment. Further, these findings are combined with a consideration of characteristics of the extraverted personality, which requires dominance and enjoys challenging and changing environments, in order to conclude that goal clarity and process clarity are the right contexts to influence the functioning of the extraversion. The findings of this paper provide empirical evidence for the theory of situational strengths and offer implications for future research.

The third theoretical contribution of this study is the extension of the boundaries of the influence mechanisms of extraverted leadership. Current research on leader extraversion has been more concerned with extraversion as a contextual moderator of the relationship between leadership style and job outcomes [41-42], overlooking the contexts that influence the expression of extraverted leadership traits [8, 34]. We introduce goal clarity and process clarity of leadership as important contexts influencing the role of extraverted leadership on performance and find that the positive effect of extraversion on team performance is weakened in situations where goal clarity and process clarity are high. The findings of this paper question the assertion that extraverted leaders are always effective and provide empirical evidence for that the characteristics of extraverted leadership vary depending on the situation. 

Practical Contributions

Our findings suggest important practical implications for the organizations seeking to identify effective team leaders. In the current era of volatility, unpredictability, complexity, and ambiguity, leaders need to navigate teams to overcome challenges and achieve various business goals. The performance advantage of extraversion in managerial and leadership positions has been claimed and proven already[43-44]. In the present study, the advantage of leader extraversion on work engagement and team performance is seen to stand out when levels of goal clarity or process clarity are low. Given the importance of leader extraversion on leader effectiveness and team performance, it is crucial for organizations to identify team leaders with moderate or high levels of extraversion. 

In addition, given that extraverted leaders like to be challenged by change, it is important to note that the ability of extraverted leaders to function may change at different stages of business development. For example, in the start-up phase, an extraverted leader can make full use of his or her energy in a changing environment, while in the maturing phase, the extraverted leader's energy may be limited by the rules. Therefore, team leaders with different levels of extroversion need to be selected at different stages of business development.

Finally,this paper offers new suggestions for leadership improvement. Favoring an extraverted, transformational approach to leadership is a crucial goal for developing effective leaders. And as the conclusions of this article demonstrate, leadership power change theory and situational leadership theory are both valid. When upper management assigns tasks to extraverted leaders, they should make full use of the strengths of extraverted leaders by giving, them more freedom to make their own arrangements.

Limitations and future research

The present study has its limitations. First, the measure of the dependent variable in this study is self-reported, which is more likely to suffer from subjectivity. Future research that uses additional measures for team performance (e.g. the entire team evaluated for team performance), and more objective measures (e.g. sales revenue for sales team) where possible would add both theoretical and practical significance to the current findings. Second, we measured extraversion as a unified dimension in this study and did not consider subdimensions of extraversion when collecting data. We suggest that future research could investigate lower-order traits[7] and different aspects[17] of extraversion for more insights. Third, although our data comprised occupationally heterogenous team leaders across several industries, we cannot guarantee the generalization of findings to all occupations or all other cultures. In the future, the sample could be expanded to improve the external validity of the findings. For example, this study was conducted in a Chinese cultural context, which emphasizes modesty and understatement, and thus the data may reflect cultural differences in the impact of extraverted leadership on team performance. Future research could investigate this topic in a more culturally diverse context and could also attempt to explore the role of different cultural values in the relationship between leadership extroversion and team performance.”

Reviewer 2:

1. I found sentence structure problems in your text that must be corrected.

Response to the Reviewer 2-1: 

Many thanks for the suggestions! We carefully proofread the whole text and revised the errors found.

In introduction part:

2. the arguments of your introduction do not arrive at your research questions.

4. Proper attention must be given to gap identification, problem, and research questions in your study

5. Research gap for mediation and moderation role must also be discussed.

6. Conceptual model of the study must be discussed in terms of research gap in existing studies and why it is important.

Response to the Reviewer 2 - 2、4、5、6: 

Thank you very much for the comments！In accordance with your comment2、comment4、comment5 and comment6, we have enhanced discourse on research gaps, reorganised contributions and rewritten the introduction. The details are as follows:

“Extraverted personality has long been acclaimed as the most effective personality variable for predicting leadership[1]. Within organizations, leaders influence others through interactions to motivate them to reach team and organizational goals. This requires leaders to possess characteristics consistent with extraversion: sociablility, contagiours energy, assertiveness, and gregariousness [2-5 ]. Two seminal meta-analytic research papers[1,6] have established that of the five-factor model (FFM) of personality traits, extraversion is the best and most consistent predictor of important leadership outcomes (i.e., leader emergence, leadership effectiveness, and transformational leadership). Furthermore, one most recent meta-analysis [7] focusing solely on extraversion provides more detailed and in-depth findings showing that extraversion and its lower-order traits are consistent predictors of leader emergence, leader behaviors (for both transformational and transactional leadership), and leadership effectiveness. Compared to other traits, thus, extraverted personality has an advantage in predicting leadership and performance.

Despite its obvious advantages for working environments, there has been limited research to date on the mechanisms underlying the effects of extraverted leadership on performance [8]. Research on how extraverted leadership affects team performance and the circumstances in which the effects of extraverted leadership are inhibited or enhanced remains limited. Judge et al [1] called for further research to focus on processes of influence of leadership traits and contexts in which leadership traits have an effect. In particular, the mechanisms of influence of the Big Five personality traits are directly related to the entire field of leadership traits research [1,4]. This article, therefore, explores the process of influencing team performance and its contextual variables in terms of extraverted personality.

In this article, we draw on earlier work on the associations between leader extraversion and team performance and proposed our hypotheses from a theoretical perspective that was different from those prior works. Drawing on the distal-proximal motivation theory[9] and the situational strength theory [10], we develop a moderated mediation model (Fig 1) that helps explain both how leader extraversion affects team performance and when such effects are suppressed or strengthened. This study makes three theoretical contributions to leader trait literature. First, we seek to enhance understanding of leader extraversion in team performance. The extant research on this topic[8.11] has drawn on dominance complementarity theory to explain this relationship and has called for more research to investigate the motivational mechanisms linking leader extraversion and team performance. Our study, responding to this call,clarifies these mechanisms using distal-proximal motivation theory. Second,we examine the effects of situational characteristics (i.e., clarity) on the expression of leader extraversion. This second objective echoes the need for further research on how individual leader differences are integrated with situational characteristics[10,12-13] .Third, we explain goal clarity and process clarity and their effect on leadership,propose a new perspective on their mechanisms in the field of leadership,and enrich the existing research on goal clarity and process clarity. From the perspective of practice, this study also provides suggestions for the selection, promotion and training of team leaders.  ”

In introduction part:

3. There are various studies on leader characteristics and team performance how your study is different from them?

Response to the Reviewer 2 - 3: 

Many thanks for the suggestions! Previous empirical studies on leadership mediating mechanisms have focused more on how leaders change the psychological states of their subordinates or team members [41], and less on changes in leaders' own psychological states. There have been calls for exploring the role of workplace engagement in the relationship between traits and performance [17].This paper intends to demonstrate that leaders' work engagement is a mediating mechanism in the relationship between leaders' extraverted personality and team performance.

In materials and methods Part:

7. Please provide the justification for using this measure of dependent variable - self-report. What are other available ways and why this one is better measure?

Response to the Reviewer 2 - 7: 

Many thanks for the suggestions! As you pointed out, and as we wrote in the limitations“Since self-reports tend to be subjective, additional measures of team performance (e.g., the entire team evaluation of team performance) and more objective measures (e.g., sales revenue for sales teams) would enhance both the theoretical and practical significance of the current findings.”However, self-report of team performance by leaders is widely used. Table 1 shows that the studies evaluate team performance by using self-report.

Table 1 Literature collation

No. Literature 

1 Bieńkowska A, Koszela A, Sałamacha A, et al. COVID-19 oriented HRM strategies influence on job and organizational performance through job-related attitudes[J]. Plos one, 2022, 17(4): e0266364.

2 Martins L L, Schilpzand M C, Kirkman B L, et al. A contingency view of the effects of cognitive diversity on team performance: The moderating roles of team psychological safety and relationship conflict[J]. Small Group Research, 2013, 44(2): 96-126.

3 Cole M S, Walter F, Bruch H. Affective mechanisms linking dysfunctional behavior to performance in work teams: a moderated mediation study[J]. Journal of Applied Psychology, 2008, 93(5): 945.

4 Kearney E, Gebert D. Managing diversity and enhancing team outcomes: the promise of transformational leadership[J]. Journal of applied psychology, 2009, 94(1): 77.

5 Zellmer-Bruhn M, Gibson C. Multinational organization context: Implications for team learning and performance[J]. Academy of management journal, 2006, 49(3): 501-518.

8. In discussion part, the discussion of findings is poorly written. It would be better if you discuss the novelty of the findings and its relevant insights with divergent views that may differ from other studies but helpful to your readers.

9. Also update the discussion part with recommendations and insights from recent literature.

10. The conclusion of the study is very shallow and merely repeating results. What are the insights of this study and what are its practical implications to real world.

Response to the Reviewer 2 - 8、9、10: 

Many thanks for the suggestions! As you pointed out, our discussion in this section is not deep and sufficient. We have rewritten this section with the following details.

“Theoretical contributions 

This study’s primary theoretical contribution is providing an empirical explanation of how leader extraversion influences team performance by acting as an actionable motivational mechanism for leader work engagement, enriching previous research on the mechanisms of extraverted leadership influence on team performance. Scholars have called for research that explores the influence processes of leadership traits [1,4].The mechanisms of influence on extraverted leadership-the type of leadership trait that has received the most attention [1]-are not clear. In this study, following Kanfer's [18] model of motivational states linking personality and work outcomes, work engagement as a motivational variable was found to be a mediating variable between extraversion and team performance. In particular, we look at leader work engagement to further enrich the study of team performance antecedent variables and leadership with the perspective of leaders' motivational states.

The second key contribution of this study is the finding that goal clarity and process clarity limit the role of extraverted leader personality, providing empirical evidence for the validity of situational strength theory. Situational strength theory suggests that strong situations can limit the role of personality in predicting individual behavior and outcomes [27], and Judge et al [10] suggest that applying this theory to discussion of the limitations of situations on the role of personality requires a clear understanding of what the situation is and how it affects personality. Roberts et al. [40] point out that it is important to find situations that are truly relevant to the role of personality. To this end, this paper looks at four aspects of the concept of situational strength and identifies goal clarity and process clarity as important factors in the work environment. Further, these findings are combined with a consideration of characteristics of the extraverted personality, which requires dominance and enjoys challenging and changing environments, in order to conclude that goal clarity and process clarity are the right contexts to influence the functioning of the extraversion. The findings of this paper provide empirical evidence for the theory of situational strengths and offer implications for future research.

The third theoretical contribution of this study is the extension of the boundaries of the influence mechanisms of extraverted leadership. Current research on leader extraversion has been more concerned with extraversion as a contextual moderator of the relationship between leadership style and job outcomes [41-42], overlooking the contexts that influence the expression of extraverted leadership traits [8, 34]. We introduce goal clarity and process clarity of leadership as important contexts influencing the role of extraverted leadership on performance and find that the positive effect of extraversion on team performance is weakened in situations where goal clarity and process clarity are high. The findings of this paper question the assertion that extraverted leaders are always effective and provide empirical evidence for that the characteristics of extraverted leadership vary depending on the situation. 

Practical Contributions

Our findings suggest important practical implications for the organizations seeking to identify effective team leaders. In the current era of volatility, unpredictability, complexity, and ambiguity, leaders need to navigate teams to overcome challenges and achieve various business goals. The performance advantage of extraversion in managerial and leadership positions has been claimed and proven already[43-44]. In the present study, the advantage of leader extraversion on work engagement and team performance is seen to stand out when levels of goal clarity or process clarity are low. Given the importance of leader extraversion on leader effectiveness and team performance, it is crucial for organizations to identify team leaders with moderate or high levels of extraversion. 

In addition, given that extraverted leaders like to be challenged by change, it is important to note that the ability of extraverted leaders to function may change at different stages of business development. For example, in the start-up phase, an extraverted leader can make full use of his or her energy in a changing environment, while in the maturing phase, the extraverted leader's energy may be limited by the rules. Therefore, team leaders with different levels of extroversion need to be selected at different stages of business development.

Finally,this paper offers new suggestions for leadership improvement. Favoring an extraverted, transformational approach to leadership is a crucial goal for developing effective leaders. And as the conclusions of this article demonstrate, leadership power change theory and situational leadership theory are both valid. When upper management assigns tasks to extraverted leaders, they should make full use of the strengths of extraverted leaders by giving, them more freedom to make their own arrangements.

Limitations and future research

The present study has its limitations. First, the measure of the dependent variable in this study is self-reported, which is more likely to suffer from subjectivity. Future research that uses additional measures for team performance (e.g. the entire team evaluated for team performance), and more objective measures (e.g. sales revenue for sales team) where possible would add both theoretical and practical significance to the current findings. Second, we measured extraversion as a unified dimension in this study and did not consider subdimensions of extraversion when collecting data. We suggest that future research could investigate lower-order traits[7] and different aspects[17] of extraversion for more insights. Third, although our data comprised occupationally heterogenous team leaders across several industries, we cannot guarantee the generalization of findings to all occupations or all other cultures. In the future, the sample could be expanded to improve the external validity of the findings. For example, this study was conducted in a Chinese cultural context, which emphasizes modesty and understatement, and thus the data may reflect cultural differences in the impact of extraverted leadership on team performance. Future research could investigate this topic in a more culturally diverse context and could also attempt to explore the role of different cultural values in the relationship between leadership extroversion and team performance.”

11. How can you enhance the generalizability of your study? Kindly work on this.

Response to the Reviewer 2 - 11: 

Thank you for pointing out the problem! The main limitation regarding generalizability is that this study was conducted in a Chinese cultural context, which emphasises low-key and humble, and that there may be cultural differences in the impact of extroverted leadership on team performance. We have strengthened this part of the discussion in the limitations and future research.

“Third, although our data comprised occupationally heterogenous team leaders across several industries, we cannot guarantee the generalization of findings to all occupations or all other cultures. In the future, the sample could be expanded to improve the external validity of the findings. For example, this study was conducted in a Chinese cultural context, which emphasizes modesty and understatement, and thus the data may reflect cultural differences in the impact of extraverted leadership on team performance. Future research could investigate this topic in a more culturally diverse context and could also attempt to explore the role of different cultural values in the relationship between leadership extroversion and team performance.”

.

---

## [Decision Letter · Decision Letter 1]

23 Nov 2022

Leader Extraversion and Team Performance: A Moderated Mediation Model

PONE-D-22-19951R1

Dear Dr. Zhang,

We’re pleased to inform you that your manuscript has been judged scientifically suitable for publication and will be formally accepted for publication once it meets all outstanding technical requirements. We also sugget you to proofread your manuscript. One of our reviewers are not satisfied with this.  

Kind regards,

Mingyue Fan, Ph.D.

Academic Editor

PLOS ONE

Additional Editor Comments (optional):

Reviewers' comments:

Reviewer's Responses to Questions

**Comments to the Author**

1. If the authors have adequately addressed your comments raised in a previous round of review and you feel that this manuscript is now acceptable for publication, you may indicate that here to bypass the “Comments to the Author” section, enter your conflict of interest statement in the “Confidential to Editor” section, and submit your "Accept" recommendation.

Reviewer #1: All comments have been addressed

Reviewer #2: All comments have been addressed

2. Is the manuscript technically sound, and do the data support the conclusions?

Reviewer #1: Yes

Reviewer #2: Yes

3. Has the statistical analysis been performed appropriately and rigorously? 

Reviewer #1: Yes

Reviewer #2: Yes

4. Have the authors made all data underlying the findings in their manuscript fully available?

Reviewer #1: Yes

Reviewer #2: Yes

5. Is the manuscript presented in an intelligible fashion and written in standard English?

Reviewer #1: No

Reviewer #2: Yes

6. Review Comments to the Author

Reviewer #1: Congratulations to all the authors for defending all the comments. However, I would suggest Language Editing.

Reviewer #2: My Decision is Accept the article "Leader Extraversion and Team Performance: A Moderated Mediation Model"

7. PLOS authors have the option to publish the peer review history of their article (what does this mean?). If published, this will include your full peer review and any attached files.

Reviewer #1: No

Reviewer #2: No

---

## [Editor Report · Acceptance letter]

1 Dec 2022

PONE-D-22-19951R1 

Leader Extraversion and Team Performance: A Moderated Mediation Model 

Dear Dr. Zhang:

I'm pleased to inform you that your manuscript has been deemed suitable for publication in PLOS ONE. Congratulations! Your manuscript is now with our production department. 

Kind regards, 

on behalf of

Dr. Mingyue Fan 

Academic Editor

PLOS ONE